# Combining Transcranial Magnetic Stimulation and Deep Brain Stimulation: Current Knowledge, Relevance and Future Perspectives

**DOI:** 10.3390/brainsci13020349

**Published:** 2023-02-18

**Authors:** Valentina D’Onofrio, Nicoletta Manzo, Andrea Guerra, Andrea Landi, Valentina Baro, Sara Määttä, Luca Weis, Camillo Porcaro, Maurizio Corbetta, Angelo Antonini, Florinda Ferreri

**Affiliations:** 1Padova Neuroscience Center (PNC), University of Padova, 35129 Padova, Italy; 2IRCCS San Camillo Hospital, Via Alberoni 70, 0126 Venice, Italy; 3IRCCS Neuromed, 86077 Pozzilli, Italy; 4Department of Human Neurosciences, Sapienza University of Rome, 00185 Rome, Italy; 5Academic Neurosurgery, Department of Neurosciences, University of Padova, 35128 Padova, Italy; 6Department of Clinical Neurophysiology, Kuopio University Hospital, University of Eastern Finland, 70211 Kuopio, Finland; 7Parkinson’s Disease and Movement Disorders Unit, Department of Neuroscience, Centre for Rare Neurological Diseases (ERN-RND), University of Padova, 35128 Padova, Italy; 8Department of Neuroscience, University of Padova, 35128 Padova, Italy; 9Institute of Cognitive Sciences, and Technologies (ISTC)-National Research Council (CNR), 00185 Rome, Italy; 10Centre for Human Brain Health, School of Psychology, University of Birmingham, Birmingham B15 2TT, UK; 11Unit of Neurology, Unit of Clinical Neurophysiology, Study Center of Neurodegeneration (CESNE), Department of Neuroscience, University of Padova, 35128 Padova, Italy; 12Venetian Institute of Molecular Medicine, 35129 Padova, Italy; 13Department of Neurology, Washington University, St. Louis, MO 63108, USA; 14Department of Neuroscience, Washington University, St. Louis, MO 63108, USA

**Keywords:** deep brain stimulation (DBS), transcranial magnetic stimulation (TMS), neuromodulation, neuropsychiatric disorders

## Abstract

Deep brain stimulation (DBS) has emerged as an invasive neuromodulation technique for the treatment of several neurological disorders, but the mechanisms underlying its effects remain partially elusive. In this context, the application of Transcranial Magnetic Stimulation (TMS) in patients treated with DBS represents an intriguing approach to investigate the neurophysiology of cortico-basal networks. Experimental studies combining TMS and DBS that have been performed so far have mainly aimed to evaluate the effects of DBS on the cerebral cortex and thus to provide insights into DBS’s mechanisms of action. The modulation of cortical excitability and plasticity by DBS is emerging as a potential contributor to its therapeutic effects. Moreover, pairing DBS and TMS stimuli could represent a method to induce cortical synaptic plasticity, the therapeutic potential of which is still unexplored. Furthermore, the advent of new DBS technologies and novel treatment targets will present new research opportunities and prospects to investigate brain networks. However, the application of the combined TMS-DBS approach is currently limited by safety concerns. In this review, we sought to present an overview of studies performed by combining TMS and DBS in neurological disorders, as well as available evidence and recommendations on the safety of their combination. Additionally, we outline perspectives for future research by highlighting knowledge gaps and possible novel applications of this approach.

## 1. Introduction

Transcranial Magnetic Stimulation (TMS) is a non-invasive neurophysiological technique that enables the investigation of the functional state of the cerebral cortex [1,2,3,4]. TMS consists of the delivery of brief, high-intensity magnetic pulses over the scalp that induce electrical currents in the superficial layers of the underlying cortex [2,4,5]. Specific paired-pulse TMS protocols, such as Short-Interval Intracortical Inhibition (SICI) and Intracortical Facilitation (ICF), can be used to test intracortical inhibitory and facilitatory circuits [3,6]. Other TMS protocols performed by pairing peripheral electrical and TMS stimuli at different interstimulus intervals (ISIs), such as Short-Latency afferent Inhibition (SAI) and Long-latency Afferent Inhibition (LAI), allow investigation of the sensorimotor integration at the cortical level [7,8,9]. Moreover, both the application of repetitive (repetitive TMS- rTMS) and patterned TMS stimuli and the repetitive coupling of TMS stimuli with peripheral electrical stimulation (Paired Associative stimulation—PAS) can induce inhibitory and facilitatory changes in cortical excitability outlasting the stimulation, thus producing synaptic plasticity [10,11]. 

In addition to stimulating superficial cortical layers, TMS has been shown to determine a more widespread modulation of cortico–subcortical networks [12,13,14]. Thus, its application may potentially induce changes in the activity and connectivity patterns of different nodes of cortico–basal–thalamic networks [15]. Hence, TMS can also represent a useful research tool in patients treated with Deep Brain Stimulation (DBS), namely, the electrical stimulation of brain structures (e.g., basal ganglia and thalamus) with surgically implanted intracranial electrodes. DBS is currently approved as an advanced treatment for different neurological conditions, including movement disorders (Parkinson’s disease (PD), essential tremor, and dystonia) [16] and epilepsy [17,18]. In addition to the delivery of the stimulation, intracranial DBS electrodes also allow the activity of the target deep brain structures in these patients to be recorded [19]. This activity may be recorded as local field potentials (LFPs), i.e., as electrical potentials generated in the extracellular space by populations of neurons surrounding the DBS lead [20,21]. 

In this context, the combination of TMS and DBS may be a useful approach to simultaneously investigate cerebral cortex and deep brain structures, as well as their mutual cortico–subcortical connections (Figure 1). Several neurophysiological and neuroimaging studies in humans and animal models showed a DBS-induced modulation of the cortex and of brain networks that has been related to the clinical effect of the stimulation [22,23,24,25]. Thus, the TMS–DBS combination may help to investigate the effects of DBS on cortical circuits, potentially revealing useful information on the mechanisms underlying its effect in different neurological conditions. This can be achieved with TMS protocols investigating cortical excitability and/or inducing cortical synaptic plasticity. On the other hand, the combination of TMS and DBS may offer the opportunity to directly investigate the effects of TMS on deep brain structures [26]. Despite its great potential, only a limited number of studies applied TMS protocols in patients treated with DBS [27,28]. This might be related to safety concerns about potential damage to brain structures and DBS electrical components [29]. 

In the present review, we first outline the results of previous studies reporting the combination of TMS and DBS in different neurological disorders. Then, following a brief report about evidence and recommendations on safety issues, we examine and discuss the potential benefits of this combined approach. Finally, we mention the current knowledge gaps in the field, highlighting some possible perspectives for future research.

## 2. Combined TMS–DBS Application: Current Evidence and Experimental Relevance

### 2.1. Evaluation of DBS Effects on Cerebral Cortex in Different Neurological Conditions

#### 2.1.1. Parkinson’s Disease

To date, most of the experimental studies applying TMS in patients treated with DBS aimed to provide insights into the mechanisms underlying DBS, improving the understanding of its effects [26,30,31]. For a number of reasons, this application has been mainly focused on the investigation of the primary motor cortex (M1) [32]. Previous studies in PD patients showed some neurophysiological abnormalities in M1, including altered excitability and plasticity [33,34,35], and these abnormalities have been suggested to play a role in the pathophysiology of cardinal motor symptoms in PD [36,37,38]. Moreover, from a methodological perspective, TMS applied over M1 induces a well-known activation of the motor system and a measurable outcome, i.e., the motor evoked potential (MEP) [4]. Unlike the stimulation of other cortical areas, MEPs can be used for the reliable, albeit indirect, evaluation of M1 neurophysiological changes. As a result, most of the available evidence comes from experimental studies in PD patients treated with DBS of the Subthalamic Nucleus (STN) or the Globus Pallidus internus (GPi), with TMS applied over M1 [26,38] (Table 1). 

In these studies, the stimulation itself was not associated with changes in motor thresholds [39,40,41,42] and in the input/output curve [39,40,43]. In addition, these measures of global cortical excitability did not change before and shortly after DBS surgery [43,44]. In addition, several studies investigated the effect of DBS on the silent period (SP) in PD. SP represents the interruption of voluntary muscle contraction induced by TMS stimuli delivered over the contralateral M1, and it is mainly mediated by intracortical inhibitory mechanisms [45,46]. A shorter SP has been consistently shown in PD patients when compared to healthy controls [47,48], suggesting a decreased activity of inhibitory circuits in M1. One study reported a shortening of a prolonged SP during GPi stimulation [39], thus suggesting that GPi-DBS could affect the intracortical inhibitory circuits in M1 responsible for SP. However, SP was longer in PD patients with the stimulator switched off with respect to healthy controls, thus complicating the generalization of the results. Studies in patients treated with STN-DBS showed more variable results. Däuper et al. [41] described a significant lengthening of the SP in STN-DBS patients when DBS was switched on, suggesting the hyperactivity of cortical inhibitory circuits mediating SP. Conversely, two other studies did not report any change in SP [40,49].
brainsci-13-00349-t001_Table 1Table 1Studies using TMS to assess DBS effects on neurophysiological measures. In all the reported studies, TMS was applied over the primary motor cortex (M1).ReferenceSubjects (n °)DBS TargetTime after DBS DBSMedicationTMS Measures (ISI)ResultsOFFONOFFONParkinson’s DiseaseChen et al. 2001 [39]PD (7)HS (7)GPi(5 unil *, 2 bil)1–5 y–++
usual med °MT, SP, I/O curve SICI (2 ms), ICF (10 ms), LICI (50/100/150/200 ms)- No change in MT, I/O curve, SICI, ICF, and LICI- SP lengthening in DBS-OFF with respect to DBS-ON and HSCunic et al. 2002 [40]PD (9)HS (8)STN(7 bil, 2 unil *)1 mo–2.7 y++
+MT, SP, I/O curve, SICI (2 ms), ICF (10 ms), LICI (50/200 ms)- No change in MT, SP, I/O curve, LICI, and ICF- Increased SICI inhibition during DBS-ONDäuperet al. 2002 [41]PD (8)HS (10)STN(bil)>3 mo++++RMT, SP, MEP latency, ICI (3 ms), ICF (13 ms)- No change in RMT, ICF, and MEP latency- Longer SP and increased ICI in DBS-ON/MED-ON with respect to DBS-OFF/MED-OFFSailer et al. 2007 [42]PD(7)HS(7)STN(bil)≈6 mo–4 y++++SAI (23 ms), LAI (200 ms)- No change in MT- SAI reduced in MED-ON/STIM OFF and increased in DBS-ON/MED-ON.- LAI increased in STIM-ON/MED-ON, reduced in STIM-OFF/MED-ON and in MED-OFF (in both DBS conditions). Hidding et al. 2006 [43]PD (8)STN(bil)Before, 2–3 w after surgery +
++RMT, I/O curve, MEP latency, maximal M response- No changes in RMT, I/O curve after surgery- Shortened MEP latencies at rest, but unchanged during muscle activation after surgery Baümer et al. 2009 [44]PD (10)STN(bil)Before, 4 d after surgery+++
RMT, SP- No change in SP and RMTKim et al. 2015 [49]PD (8)HS (9)STN(bil)≈9 mo–9 y++++MT, SPPAS (21.5 ms)- No change in MT and SP- MEP size increase in DBS-ON condition- Increased PAS response in PD only in DBS-ON/MED-ON condition. Casula et al. 2016 [50]PD (6)HS (8)STN(bil)3–10 y++++RMT, TMS-evoked potentials, TMS-evoked spectral perturbation-No change in RMT- Modulation of early TMS-evoked activity by DBS stimulation- Additional modulation of later TMS-evoked components induced by DBS stimulation and L-Dopa**Dystonia**Kühn et al. 2003 [51]DYST(9)HS(20)GPi (8 bil, 1 unil, 4 with VIM)3–11 mo++Medicated and non-medicatedRMT, SP, ICI (1–7 ms), ICF (10/15 ms), SRC - No change in ICI and ICF- Higher MT in DBS OFF with respect to DBS-ON- Higher SRC with DBS-ON Tisch et al. 2007 [52]DYST(10)HS(10)GPi(bil)>6 mo++Medicated and non-medicatedMT, SRCPAS (25 ms)- No change in MT- Reduced PAS response in DBS ON Ruge et al. 2011 [53]DYT1 DYST(10)HS(10) GPi(bil)4.5–11.5 y++Not clearly reported °MT, I/O curve, SICI (2/3 ms)- No change in MT, I/O, and SICI- No change in PAS response with DBS- Reduced PAS response in DYST patientsRuge et al. 2011 [54]Primary DYST(8)GPi(bil)Before and 1–3–6 mo after surgery 
+Not clearly reported °MT, I/O curve, SICI (2/3 ms)PAS (25 ms)- No change in MT and I/O curve-SICI reduced before surgery, then progressively increased after surgery- Increased PAS response before surgery, absent after 1 mo, then increased at 3 and 6 moWagle Shukla et al. 2018 [55]Cervical DYST(10)HS(10)STN(bil)1–6 y++Medicated and non-medicatedRMT, SICI 2/3 ms, ICF 10/15 ms, SAI 20/30 ms, LAI 150/200 msPAS (25 ms)- No change in RMT, SICI and ICF, LAI 150 ms- In DBS-ON increased SAI, LAI 200 ms and reduced PAS response with respect to DBS-OFF**Essential tremor**Molnar et al. 2005 [56]ET(7)HS(11)VIM(unil)3.1–7.5 y++Non-medicatedMT, SP, I/O curve, SICI (2 ms), ICF (10 ms), LICI (50/100/150/200 ms)- No change in MT, SP, SICI, ICF, and LICI- Increased MEP amplitude during DBS-ON for TMS intensities >130% RMT (I/O curve) Molnar et al. 2004 [57]ET(6)HS(9)VIM(unil)1.2–7.5 y++ Non-medicatedPaired cerebellar and M1 TMS stimulation (from 3 to 7 ms)- No change in MEP amplitude with DBS- With paired stimulation reduced MEP amplitudes in DBS-ON at ISIs of 6–7 ms. **Epilepsy**Molnar et al. 2006 [58]Epilepsy(5)HS(9)ANT(bil)Not clearly reported++AEDMT, SP, I/O curve, SICI (2 ms), ICF (10 ms), LICI (50/100/150/200 ms)- No change in MT, SP, I/O curve, ICF, and LICI- Reduced SICI with DBS-ON (similar to HS)* In the study by Cunic et al. [40], one patient had unilateral DBS-STN, while another patient had unilateral DBS-STN and contralateral pallidotomy. In the study by Chen et al. [39], patients with unilateral GPi-DBS had contralateral pallidotomy. ° The authors do not specify patients’ medication, but report that medication was not changed during the study. Abbreviations: AED = antiepileptic drugs; ANT = anterior nucleus of the thalamus; bil = bilateral DBS implantation; DYST = dystonia; DBS = deep brain stimulation; ET = essential tremor; GPi = Globus Pallidus internus; HS = healthy subjects; ICF = intracortical facilitation; ISI = inter-stimulus interval; LAI = long-latency afferent inhibition; LICI = long-interval intracortical inhibition; MEP = motor-evoked potential; MT = motor threshold (refers to active motor threshold + RMT measurement); PAS = paired associative stimulation; PD = Parkinson’s disease; RMT = resting motor threshold; SAI = short-latency afferent inhibition; SICI = short-interval intracortical inhibition; SP = silent period; SRP = stimulus–response curve; STN = subthalamic nucleus; unil = unilateral DBS implantation; VIM = ventral intermediate nucleus of the thalamus.


Moreover, some studies in PD patients investigated the effects of DBS on SICI, a paired-pulse TMS measure reflecting the activity of GABA-A-ergic intracortical circuits [59,60], which is increased in PD patients [6,33,61]. In patients treated with STN-DBS, SICI decreased when DBS was switched on [40,41,44,62,63] with respect to the switched off condition. This suggests that STN-DBS may restore inhibitory mechanisms underlying SICI in M1. Conversely, another study on GPi-DBS did not find any significant effect of the DBS stimulation on SICI [39]. Neither STN-DBS nor GPi-DBS produced any change in ICF, reflecting the activity of glutamatergic intracortical circuits [39,40,41]. 

The effects of DBS on SAI and LAI in PD patients were investigated in a single study, where modulation of both measures with STN-DBS was reported [42]. In more detail, SAI and LAI were both reduced in PD patients after the administration of L-Dopa when the stimulator was switched off and switching on the DBS normalized both measures. LAI was reduced in patients not taking L-Dopa, and DBS stimulation did not induce any change in this measure. These results show that DBS can modulate afferent inhibition and that this effect is influenced by L-Dopa.

Moreover, multimodal approaches combining TMS-DBS with neurophysiological techniques for the study of cortical activity, such as electroencephalography (EEG) and magnetoencephalography, represent a promising tool to investigate the effect of DBS stimulation on cortical areas. To date, only one study has evaluated the effects of STN-DBS in PD patients using a TMS–EEG approach [50], showing that invasive stimulation can effectively modulate cortical activity. 

Hence, based on the available evidence, DBS targeting STN and GPi can modulate different inhibitory intracortical circuits in M1, as well as sensorimotor integration at the cortical level in PD patients, as measured by different TMS protocols. The observed differences in SICI and SP between patients treated with STN-DBS and GPi-DBS have mainly been related to the different therapeutic effects of the stimulation [26,38]. However, the evidence regarding the modulation of intracortical circuits by GPi-DBS comes from a single study, while the heterogeneity of experimental methods and studied populations must also be considered. Thus, mechanisms mediating the observed results remain controversial. 

#### 2.1.2. Dystonia

Studies in patients with dystonia treated with GPi-DBS showed a DBS-induced modulation of M1 excitability and intracortical inhibitory circuits. 

A first study by Kühn et. al. [51], reported a decrease in resting motor threshold and an increase in the steepness of the MEP recruitment curve during GPi stimulation, suggesting a global increase in cortical excitability when the DBS was switched on. However, later studies did not confirm these findings [52,53,54,64] (Table 1).

In patients with dystonia, a decreased SICI has been described by several studies, reflecting reduced inhibition within the motor cortex [65,66,67]. Two studies assessed whether this neurophysiological abnormality could be reversed in chronically-implanted patients by GPi-DBS [51,53]. SICI was tested with the DBS switched both off and on, but no change was found between the two experimental conditions. These results suggest the lack of acute positive effects of GPi-DBS on impaired M1 inhibition in dystonia. However, in a longitudinal study SICI was assessed before and after DBS surgery, and altered SICI progressively improved after 1, 3 and 6 months following DBS implantation, mirroring the time course of clinical symptoms [54]. The same study also described the time course of PAS stimulation at the same intervals, which will be discussed later. Sensorimotor integration tested by SAI and LAI has never been investigated in patients with dystonia and GPi-DBS, while one study in patients with cervical dystonia and STN-DBS described a positive effect of stimulation on both SAI and LAI 200 ms [55] (Table 1). 

The major findings of studies applying TMS and DBS alone, as well as their combined application in patients with dystonia, have been discussed in detail in a recent review by Udupa et al. [28].

#### 2.1.3. Essential Tremor

To date, only two studies have been performed to explore the effect of DBS of the Ventral Intermediate nucleus (VIM) of the thalamus in patients with essential tremor (ET) using a combined TMS-DBS approach (Table 1). By applying TMS over M1 in different DBS stimulating conditions, Molnar and colleagues [56] described an increase in M1 global excitability when the DBS was switched on compared to the switched off condition, thus suggesting that DBS effects on ET could result from the facilitation rather than inhibition of VIM activity. 

In a previous study by the same group [57], patients treated with unilateral DBS targeting the VIM underwent a paired cerebellum and M1 TMS stimulation, applied at different ISIs (from 3 to 7 ms) to assess the cerebello–thalamo–cortical pathway. Switching on the DBS resulted in an increase in cortical excitability and thus in the facilitation of the cerebello–thalamo–cortical pathway, thus further pointing to a possible DBS activation of the VIM in ET patients treated with DBS [57]. 

#### 2.1.4. Epilepsy

To date, only one TMS study was performed in a small group of patients with drug-resistant epilepsy treated with bilateral DBS implantation in anterior nuclei of the thalamus (ANT) [58]. In these patients, DBS stimulation induced an increase in SICI in M1, thus indicating a modulation of M1 cortical inhibitory circuits (Table 1). However, the small sample size and the heterogeneous characteristics of the included patients do not allow conclusions to be drawn on the effects of DBS on cortical excitability in epileptic patients treated with DBS of the ANT, and more evidence is needed to confirm this finding. However, the risk of inducing epileptic seizures should be further investigated as a potential limitation for the application of TMS in these patients [29] (see the section on safety issues below).

#### 2.1.5. Overview

Overall, these studies provide relevant information on abnormalities in cortical excitability, connectivity, and plasticity in several neurological conditions, expanding our knowledge of their pathophysiology. More research should be conducted on additional neuropsychiatric conditions where DBS is emerging as a therapeutic opportunity, such as Alzheimer’s disease and other dementias.

### 2.2. Modulation of Cortical Synaptic Plasticity

#### 2.2.1. Induction of Cortical Plasticity with TMS in Patients with DBS

Another potential experimental approach is the application of patterned TMS protocols to assess if DBS stimulation can modulate cortical synaptic plasticity. This could be particularly important in the study of neurological disorders with well-described abnormalities in several types of experimentally induced cortical plasticity, such as PD and dystonia [28,68,69]. In fact, converging evidence indicates that abnormal cortical synaptic plasticity plays a role in the pathophysiology of these disorders [70,71]. Several TMS paradigms, such as PAS and rTMS, can be used to induce cortical plasticity. When these protocols are applied to the M1, the induction of plasticity can be evaluated with post-intervention changes in MEP amplitudes outlasting brain stimulation [72,73]. 

The induced plasticity shares several features with associative long-term potentiation and long-term depression described in the hippocampus, such as its associativity, long duration, and interaction with motor learning [72,74,75].

RTMS consists of the delivery of repetitive magnetic stimuli that induce persistent inhibitory or facilitatory changes in cortical excitability depending on the frequency and the pattern of the stimuli [73].

PAS consists of the pairing of peripheral electrical stimuli of the median nerve with single TMS pulses applied over the contralateral M1 at specific time intervals and induces spike-timing-dependent plasticity in the motor cortex [72,74,76]. PAS-induced changes in MEP amplitudes depend on the interval between the electrical and the magnetic stimuli. Specifically, an enhancement of cortical excitability can be induced using a 25 ms interval, while an interval of 10 ms results in inhibitory changes in the cortical excitability of M1 [74]. However, the effects of rTMS and PAS are usually limited in duration, and the clinical benefit in neurological disorders such as PD and dystonia is questioned [72]. The combination of these techniques with DBS could produce more consistent changes in synaptic plasticity with potential therapeutic benefits. 

Although the mechanisms underlying DBS’s effects are still unclear, the modulation of cortical and subcortical synaptic plasticity could be responsible for the clinical benefits of the stimulation. Evidence from animal models and imaging studies in humans suggest the existence of long-lasting effects on brain activity, thus supporting a DBS-induced modulation of brain plasticity [77,78]. Moreover, it is still debated whether the delayed effect of the stimulation on clinical symptoms in patients with dystonia and PD could depend on the modulation of brain plasticity [78,79]. To date, PAS is the only TMS protocol applied in DBS patients to assess changes in cortical synaptic plasticity [26,28]. 

In a study performed on PD patients treated with STN-DBS, the PAS protocol was applied with the stimulator switched on and off and with and without the acute administration of levodopa [49]. PAS induced an increase in MEP amplitudes only in patients with the stimulator switched on and taking levodopa, thus restoring the impaired M1 Long Term-Potentiation-like (LTP-like) plasticity. To the best of the author’s knowledge, to date, no study has applied the PAS protocol in PD patients treated with GPi-DBS. 

Conversely, in a group of patients with primary generalized dystonia, the PAS protocol was applied when DBS was switched on and induced a decrease in motor cortical excitability with respect to both the switched-off condition and the healthy controls [52], thus suggesting a decrease in LTP-like plasticity induced by the stimulation. Conversely, in a study in patients with cervical dystonia treated with STN-DBS, an increase in MEP amplitude was observed when the stimulator was switched off [55] with respect to the DBS-on condition and to HS. 

In another study, with the aim of evaluating changes in cortical plasticity over time, PAS was applied in dystonic patients before and 1, 3, and 6 months after GPi-DBS surgery [54], with the stimulator switched on. Before surgery, PAS-induced plasticity was increased in patients when compared to healthy controls, was ineffective at 1 month, and then increased to the levels of healthy subjects at 3 and 6 months (Table 1). These observations further point toward modulation of abnormal cortical plasticity in patients with dystonia. Because of the lack of ex vivo safety studies, rTMS has never been systematically applied in patients with DBS, and to date, only a case report in a patient with obsessive-compulsive disorder (OCD) has been published [80].

#### 2.2.2. Pairing of TMS and DBS Stimuli to Induce Cortical Synaptic Plasticity

The pairing of TMS and DBS stimuli may represent a novel and intriguing method to induce cortical synaptic plasticity. In order to produce spike-timing-dependent plasticity in the cortex, the coupling of magnetic and electrical stimuli should be performed at ISIs close to, or corresponding to, the time needed for neural signals to travel between the stimulated structures [81]. Transmission times can be estimated using cortical evoked potentials generated by DBS stimuli and recorded with EEG [63]. Only a few studies were performed to date with this paired-pulse approach, with DBS stimuli preceding single TMS pulses. 

In a first study in patients with PD, an increase in MEP amplitude was described at ISIs of 3 and 23 ms between conditioning STN-DBS stimuli and an M1 single-pulse TMS [63]. Based on studies in animal models, the early facilitation at the ISI of 3 ms has been attributed to an antidromic transmission of DBS stimuli to M1 via the hyperdirect pathway [82,83], while the later effect at 23 ms was attributed to the polysynaptic transmission via the basal–thalamic–cortical circuits [61,84,85]. 

In another study in PD patients, DBS stimuli delivered in the STN were repeatedly coupled with single TMS pulses applied over M1 at different ISIs [86]. In line with the aforementioned reports on conduction times between the STN and the motor cortex, a long-term increase in MEP amplitude was described after the paired stimulation at ISIs of 3 ms and 23 ms, with respect to a control ISI of 167 ms [86]. 

Similarly, in patients with cervical dystonia treated with bilateral DBS of the Gpi, DBS pulses were paired with single-pulse TMS at different ISIs [27]. The tested ISIs were chosen based on the peak latencies of Gpi-DBS-induced, cortically evoked potentials in M1 recorded by EEG [27]. Long-term MEP facilitation was described at ISIs of 10 and 25 ms, thus suggesting the induction of spike-dependent synaptic plasticity on the motor cortex.

### 2.3. Evaluation of TMS-Induced Changes in Deep Brain Structures

DBS–TMS coupling could represent an experimental approach to assess TMS-induced effects on the activity of basal ganglia and thalamus by recording LFPs with DBS leads. Magnetic stimuli delivered with single-pulse TMS are thought to affect the most superficial layers of the cerebral cortex [5]. Direct evidence of more widespread effects on cortico–basal–thalamic networks is still scarce [87]. Considering its direct connection with M1 via the hyperdirect pathway, the STN may represent an appropriate target for the investigation of a possible TMS-induced modulation of the activity of deep brain structures [14,88,89]. In fact, together with the modulation of local cortical circuits, the activation of cortical neurons with TMS may further spread via cortico–cortical and cortico–subcortical projections, thus involving brain networks [14]. Results of several functional neuroimaging studies and EEG recordings support this indirect modulation by TMS [15,88]. The assessment of the effect of plasticity-inducing TMS protocols (e.g., PAS and rTMS) on the activity of deep brain structures would be of particular interest. 

The recording of LFPs for deep brain structures with DBS electrodes may thus provide insights on cortico–basal connections and their contribution to DBS’s effects [26].

## 3. Safety Issues

The application of TMS in patients treated with DBS raises several safety issues regarding the potential of magnetic field pulses to induce electrical voltages and currents in DBS electrical components, such as intracranial leads, subcutaneous wires, and implantable pulse generators. In turn, this might lead to DBS system malfunctions and to potentially harmful brain damage. This topic was addressed by a first Expert Consensus in 2009 [11], promoted by the International Federation of Clinical Neurophysiology, and was more extensively discussed in a recent update based on new emerging experimental evidences [29].

To further address this issue, Magsood et al. [90] measured currents induced by TMS in a DBS probe inserted in an anatomically accurate brain phantom that mimicked conductivity properties of brain components. The results showed that the measured currents were higher than those produced by DBS. This occurred when the highest TMS intensity (100% of maximum stimulator output) was applied directly over the DBS leads, thus raising the possibility of an overstimulation during the combined application of both TMS and DBS.

Accordingly, although the application of TMS in DBS patients is overall considered a safe approach, several factors should be considered, including: (i) the position of the TMS coil with respect to the DBS leads and subcutaneous extension wires; (ii) the distance between the TMS coil and the implantable pulse generator; (iii) the presence, number, and direction of loops in extension wires. The higher risk of inducing high voltages in the DBS system is associated with TMS coil placed directly over DBS leads and the TMS-induced current (which is perpendicular to the coil itself) aligned with the DBS lead or centered over extension wire loops [29,91,92]. Concerning the distance between TMS coil and the DBS pulse generator, Kumar et al. [93] described a higher risk of system malfunction at <2 cm, while a distance between 2 and 10 cm was associated with the shut-off of the DBS device. Distances between 10 and 30 cm had no effect on the implantable pulse generator’s function.

Although the use of E-Field estimation and imaging techniques for guiding the delivery of TMS is considered a safer approach with respect to non-navigated TMS, there is currently no evidence suggesting its benefits in terms of safety in patients with DBS [29]. 

Moreover, as new DBS systems will become available, the aforementioned issues should be evaluated according to the characteristics of the stimulating device [29]. 

To date, the safety of the application of rTMS was assessed only in one patient with OCD treated with bilateral DBS of the nucleus accumbens and anterior arm of the internal capsule [80]. In their single case report, Miron et al. [80] performed low-frequency (1 Hz) and high-intensity (100% of the maximum stimulator output) rTMS of the right orbitofrontal cortex, switching off the DBS during each session. rTMS did not produce long-lasting improvements of OCD symptoms. However, the application of RTMS induced neither shifts in DBS leads nor malfunctions in the DBS system. 

Only a few studies were performed combining rTMS with other implanted electrical devices. Phielipp et al. [94], reported a single patient with chronic pain treated with the implantation of a subdural electrode placed on the right motor cortex undergoing 10 sessions of rTMS. In each session, stimulation consisted of twenty 5 s trains of 100 stimuli at 20 Hz (total of 2000 stimuli) applied at an intensity of 90% of the patient’s RMT (40 ± 2% of maximum stimulator output in different sessions) over the right motor cortex. Despite previous evidence suggesting a benefit for the treatment of chronic pain [95], the patient did not report any clinical benefit and experienced two episodes of involuntary muscle contractions. In a multicenter survey, Philip et al. [96] reported 20 patients with drug-resistant depression treated with vagal nerve stimulators undergoing rTMS. However, the authors neither reported the TMS protocols used, nor the cortical areas targeted. In both studies, DBS was switched off during the stimulation and no system malfunction was reported. 

However, despite the lack of major adverse effects, the results of these studies cannot be generalized to the application of rTMS in patients with DBS, and more evidence assessing the safety of this procedure is required.

## 4. Future Perspectives

Recent advances in DBS technology are offering new perspectives for the study of abnormal neural activity of deep brain structures in several neurological disorders and for the understanding of mechanisms underlying DBS effects [97]. Importantly, the recent advent of sensing-enabled technologies for DBS systems is offering the opportunity to directly record neural activity in deep brain structures chronically and in a non-laboratory setting [98].

As discussed above, these recordings potentially allow for the evaluation of TMS-induced effects on deep brain structures and a better characterization of connections between DBS targets and cortical areas. 

Furthermore, DBS is currently under investigation as a treatment for other neuropsychiatric disorders (e.g., Tourette syndrome, depression, and Alzheimer’s disease), and several deep brain structures, as well as cortical areas, are emerging as new potential targets [99]. As more evidence will become available, this could represent an important opportunity to evaluate the role and the function of DBS target structures in these disorders, such as thalamic nuclei and GPi in Tourette syndrome [100], limbic structures in treatment-resistant depression [101], or the fornix and the nucleus basalis of Meynert in Alzheimer’s disease [102]. In this scenario, TMS–DBS coupling may represent a valuable research tool to investigate neurophysiological changes induced by DBS on the cortex and their association with clinical improvement.

Moreover, an important issue that should be addressed concerns the reproducibility and clinical effect of TMS and DBS stimuli coupling as a plasticity-inducing method.

As discussed above, several safety concerns are currently limiting the combined TMS–DBS application [29]. Importantly, evidence supporting the safety of the application of rTMS in patients treated with DBS is still lacking, and therefore, studies with ex vivo models assessing the feasibility and safety of this approach are crucially needed. The combined application of rTMS and DBS would be of particular interest as a neuromodulation technique in psychiatric disorders where TMS has shown some therapeutical benefits, such as OCD and treatment-resistant depression [103]. This offers a background for the application of combined neuromodulation as a further therapeutic tool. Furthermore, the clinical benefit and neurophysiological effects of experimental protocols combining rTMS with operative learning in DBS patients could be investigated [104,105]. In fact, a study by Karim et al. [104] suggested that the combination of perceptual learning with rTMS induced significant after-effects that were not observed when the two protocols were used separately. 

Although OCD is an approved indication for DBS therapy [106], to date, TMS studies on OCD patients treated with DBS are lacking. One of the reasons is probably the still limited access of these patients to DBS therapy [107].

Concerning movement disorders, although DBS targeting the VIM of the thalamus is an approved treatment for PD patients with severe tremor, studies combining TMS with DBS have never been performed in these patients. 

A further potential development of the TMS–DBS combined application could be represented by multimodal approaches combining other neurophysiological techniques, such as EEG [50], magnetoencephalography, as well as functional neuroimaging [28]. 

## 5. Conclusions

Experimental studies combining TMS and DBS have provided insights into the modulation of cortical excitability and plasticity induced by DBS, particularly in movement disorders. These effects have emerged as putative mechanisms underlying DBS effects and have provided important clues on DBS’s mechanisms of action. However, combining TMS and DBS is currently limited by several safety concerns, and the application of this experimental approach is below its true potential. With the advances in DBS technology and with the identification of new targets, a deeper understanding of DBS effects will become crucial for its clinical application. Moreover, the combined neuromodulation offered by the repetitive pairing of TMS and DBS stimuli could represent a novel therapeutic approach whose clinical relevance needs to be clarified. 

## Figures and Tables

**Figure 1 brainsci-13-00349-f001:**
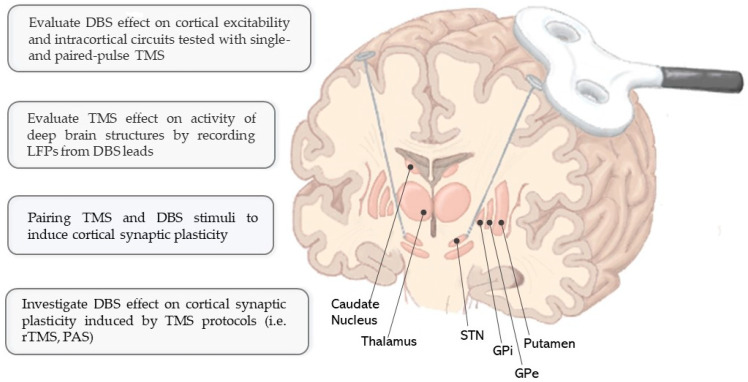
Schematic representation of TMS and DBS coupling and its potential advantages. DBS: deep brain stimulation; GPe: external globus pallidus; GPi: globus pallidus internus; LFPs: local field potentials; PAS: paired associative stimulation; TMS: transcranial magnetic stimulation; STN: subthalamic nucleus. The figure was created by V.D. with Inkscape (version 1.2.2).

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
