# Peer review of "Combining Transcranial Magnetic Stimulation and Deep Brain Stimulation: Current Knowledge, Relevance and Future Perspectives"

_brainsci, 2023, doi:10.3390/brainsci13020349_

Round 1

Reviewer 1 Report

 Combining Transcranial Magnetic Stimulation and Deep Brain Stimulation: A Review paper by

D’ Onofrio et al.

Deep brain stimulation (DBS) has been shown to be useful in the treatment of neurological disorders such as tremor and Parkinson (1) but has shown only limited effects in psychiatric disorders such as obsessive compulsive disorder (OCD) or depression (2). In contrast, transcranial magnetic stimulation (TMS) has been shown to be useful both to measure neuroplastic change as dependent variable (e.g. measuring motor evoked potentials, stimulus-response curves and silent periods) and to induce neuroplastic change by repetitive stimulation protocols as a treatment variable (3).

In their review article D’Onofrio et al. are discussing the relevance and possible clinical potentials in combining TMS with DBS.

In general, this is a well-conducted review paper on a highly interesting method which could shed light on the neurophysiological effects of TMS and DBS in understanding the pathophysiology of neurological and psychiatric disorders as well as in developing novel treatment protocols.

The introduction gives a clear overview over the field of research; methods and results of previous studies are described clearly, and adequately discussed. I have only some suggestions for improvements which are listed below:

1.    The authors clearly describe the possibility to evaluate DBS effects on cortical excitability using TMS as a dependent variable and the possibility to evaluate TMS effects on the activity of deep brain structures by recording local field potentials (LFPs) from DBS leads. However, concerning the intriguing possibility to pair rTMS with DBS both as treatment variables to induce synaptic plasticity, the authors should explain (1.) the mechanism of action of DBS (2.) the mechanism of action of rTMS and (3.) the putative neurophysiological mechanisms underlying the combination of rTMS and DBS. The main question here is: Why should the pathophysiology of patients suffering from neurological or psychiatric disorders profit from such a combination? This review paper would profit from an elaborative discussion of the neurophysiological processes induced by DBS alone, rTMS alone and the possible  effects induced by the combination of DBS and rTMS.   

2.    Page 7, line 298-300: The authors should explain the main methods and findings from their cited studies (86-88) combining rTMS with DBS. Which stimulation parameters were used and what was the rational for this choice. Which effects did this stimulation protocol have on OCD?

3.    Page 13, line 332: please change the term “neurological disorders” in “psychiatric disorders” since the authors are referring here to OCD and treatment-resistant depression.

4.    Page 13, line 327-344: The authors discuss potential clinical applications of TMS-DBS coupling with other neurophysiological techniques such as EEG, MEG and fMRI. The authors might also consider the clinical and neurophysiological benefit in combining TMS-DBS with a brain stimulation paradigm introduced by Karim et al. (4) showing that combining rTMS with operant learning can induce persistent after-effects not achievable neither with rTMS alone nor with operant learning alone. The clinical benefit of this paradigm has been shown e.g. by Khedr et al. (5) in neurorehabilitation. 

References:

1.    1. Krack, P.; Volkmann, J.; Tinkhauser, G.; Deuschl, G. Deep Brain Stimulation in Movement Disorders: From Experimental Sur-403 gery to EvidenceBased Therapy. Mov Disord 2019, 34, 1795–1810, doi:10.1002/mds.27860.

2.     2. Hyde, J.; Carr, H.; Kelley, N.; Seneviratne, R.; Reed, C.; Parlatini, V.; Garner, M.; Solmi, M.; Rosson, S.; Cortese, S.; et al. Efficacy 591 of Neurostimulation across Mental Disorders: Systematic Review and Meta-Analysis of 208 Randomized Controlled Trials. Mol Psychiatry 2022, 27, 2709–2719.

3.     3. Klomjai W, Katz R, Lackmy-Vallée A. Basic principles of transcranial magnetic stimulation (TMS) and repetitive TMS (rTMS). Ann Phys Rehabil Med. 2015, 58(4):208-213.

4.    4.  Karim AA, Schüler A, Hegner YL, Friedel E, Godde B. Facilitating effect of 15-Hz repetitive transcranial magnetic
stimulation on tactile perceptual learning. J Cogn Neurosci. 2006, 18:1577-1585.

5.     5. Khedr EM, Abo El-Fetoh N, Ali AM, El-Hammady DH, Khalifa H, Atta H, et al. . Dual-hemisphere repetitive transcranial magnetic stimulation for rehabilitation of poststroke aphasia: a randomized, double-blind clinical trial. Neurorehabil Neural Repair 2014, 28:740–50.

Author Response

Deep brain stimulation (DBS) has been shown to be useful in the treatment of neurological disorders such as tremor and Parkinson (1) but has shown only limited effects in psychiatric disorders such as obsessive compulsive disorder (OCD) or depression (2). In contrast, transcranial magnetic stimulation (TMS) has been shown to be useful both to measure neuroplastic change as dependent variable (e.g. measuring motor evoked potentials, stimulus-response curves and silent periods) and to induce neuroplastic change by repetitive stimulation protocols as a treatment variable (3).

In their review article D’Onofrio et al. are discussing the relevance and possible clinical potentials in combining TMS with DBS.

In general, this is a well-conducted review paper on a highly interesting method which could shed light on the neurophysiological effects of TMS and DBS in understanding the pathophysiology of neurological and psychiatric disorders as well as in developing novel treatment protocols.

The introduction gives a clear overview over the field of research; methods and results of previous studies are described clearly, and adequately discussed. I have only some suggestions for improvements which are listed below:

We thank the reviewer for the positive comments.

1) The authors clearly describe the possibility to evaluate DBS effects on cortical excitability using TMS as a dependent variable and the possibility to evaluate TMS effects on the activity of deep brain structures by recording local field potentials (LFPs) from DBS leads. However, concerning the intriguing possibility to pair rTMS with DBS both as treatment variables to induce synaptic plasticity, the authors should explain (1.) the mechanism of action of DBS (2.) the mechanism of action of rTMS and (3.) the putative neurophysiological mechanisms underlying the combination of rTMS and DBS. The main question here is: Why should the pathophysiology of patients suffering from neurological or psychiatric disorders profit from such a combination? This review paper would profit from an elaborative discussion of the neurophysiological processes induced by DBS alone, rTMS alone and the possible effects induced by the combination of DBS and rTMS.  

      We thank the reviewer for this suggestion, and we believe that these are important points to discuss. Accordingly, in the revised version of the manuscript, we have extensively modified section 2.2.1 following  the reviewer’s indications. Particularly, we have mentioned the pathophysiological role of synaptic plasticity in movement disorders and rTMS mechanisms of action (point 2). Regarding point 1, DBS effects on cortical and subcortical plasticity are still largely unknow and the available evidence comes from a limited number of imaging studies and from the evidence of the delayed effect of the stimulation on motor and non-motor symptoms in PD and dystonia. Putative neurophysiological mechanisms underlying the combination of rTMS and DBS are still unexplored, but we offered an hypothesis on the neurophysiological outcomes of the combination. Specifically, we speculate that rTMS and DBS combined application could result in an enhancement of the effect of both techniques on cortical synaptic plasticity.

2) Page 7, line 298-300: The authors should explain the main methods and findings from their cited studies (86-88) combining rTMS with DBS. Which stimulation parameters were used and what was the rational for this choice. Which effects did this stimulation protocol have on OCD?

      As suggested by the reviewer, we have extensively reported methods and findings of studies combining rTMS and invasive devices (page 11-12, line 385 403). We have also clarified that the study by Philip et al. [96] was a multicentric survey and rTMS parameters as well as targeted cortical areas were not reported in detail for each patient.

3) Page 13, line 332: please change the term “neurological disorders” in “psychiatric disorders” since the authors are referring here to OCD and treatment-resistant depression.

In the revised version of the manuscript, we have now replaced the term “neurological disorders” with “psychiatric disorders” (page 13, line 436)

4) Page 13, line 327-344: The authors discuss potential clinical applications of TMS-DBS coupling with other neurophysiological techniques such as EEG, MEG and fMRI. The authors might also consider the clinical and neurophysiological benefit in combining TMS-DBS with a brain stimulation paradigm introduced by Karim et al. (4) showing that combining rTMS with operant learning can induce persistent after-effects not achievable neither with rTMS alone nor with operant learning alone. The clinical benefit of this paradigm has been shown e.g. by Khedr et al. (5) in neurorehabilitation.

We thank the reviewer for the interesting suggestion. We have briefly discussed the possible usefulness of combining learning paradigms and rTMS in the paragraph concerning possible advantages in combining rTMS and DBS (page 13, lines 439-443), presenting the case of the combination of perceptual learning and rTMS proposed in the study of Karim et al.

Reviewer 2 Report

-Abstract, raw 39-42 should be rewritten clearly as the aim of the study.

-Fig 1 what is the source? Ownership of the authors?

-Raw 151-153 – please explain more in detail.

-Raw 166 – was motor threshold related to resting or active? Kuhn et al. should be written as “Kuhn et al. [Reference number] reported…”

-Raw 171-172, related to studies referenced as 57 and 63, please explain more clearly, whether the stimulation was on on or off?

-The acronym usage should be properly checked in the whole manuscript since there are inconsistencies for example: The acronym PAS should be written as the full term in the text (see raw 175, 216). Previously the PAS is introduced as a full term only in the legend of Fig1. The same is for LPF in the text (raw 270), only introduced in Fig 1 as the full term, not in the manuscript text.

Raw 279 is written with the full term “electroencephalographic” and in raw 249, the acronym EEG was introduced.  

-Please reduce the number of acronyms in the manuscript.

-Raw 180-181, please write in more detail what was done in the Udupa study, and what is the difference between the submitted manuscript and the Udupa?

-Raw 191- “in another study” please rewrite to be more understandable..

-Table 1 should be referenced properly in the text, not only in the 3.1.1 section. If  Table 1“covers” other diseases and conditions, then Table 1 referencing should be placed accordingly. It is believed the authors made a mistake with referencing 3.1.1., it should be 2.1.1., and also other paragraphs related to other conditions (dystonia, essential tremor..)?

-Thalamic nuclei should be written with full term or with introducing acronyms, there is nonconsistency in the manuscript (for example, raw 184 versus 197, etc.). It is suggested to use fewer acronyms since it will be too many for the readers to follow.

-Raw 206, please rewrite to be more understandable “” these studies shed some light..:”

-Section 2.2. should be 3.1 and, accordingly, the subheadings? Also, the title should be slightly modified, what does it mean by “study and modulation?

- Raw 225 -“has never been” please rewrite to be more acceptable

-Raw 240- r-TMS, the term is not previously introduced as full term, please reduce the acronym used. The r-TMS is usually used as rTMS.  “has never been” please rewrite to be more acceptable. Raw 242 misses the reference/references? In raw 298, the authors use the term rTMS.

-Separate paragraph devoted to technical solutions and limitations for the safety application of TMS and DBS should be given, not placing these facts in 2.3. paragraph. No discussion can be found on the difference between the TMS devices (standard TMS with stimulator and EMG device, line navigated TMS and e-field navigated TMS) in usage with DBS in respect of the safety issues or technical compatibilities.

-Raw 292 - Kumar et al. should be properly written as Kumar et al. [reference number] according to journal guidelines for text referencing.

-Raw 299, “Vagal nerve Stimulator”, why S is the capital, and why V is the capital letter?

- Cunic [40] The authors studied the effects of subthalamic nucleus (STN) deep brain stimulation (DBS) on these circuits in 12 patients with PD treated with STN DBS. Please check in Table 1  the number of patients PD (9) and HS 8? The data from HS are served as references/norms for TMS measures right?

-It would be suggested to order the studies in Table 1 according to years from more recent to older or from older to more recent studies, and also change the reference numbers accordingly in the text. Or the studies in Table should be ordered as they appeared in the manuscript text (how come study 40 is before 39 in Table 1?).

-4.1. the margin lining should be placed accordingly to journal guidelines and accordingly to the previous text, here, the paragraphs start as the margins of Table 1.

-Raw 318 – “..currently under investigation as a treatment for several other neurological disorders, and new deep brain structures as well as cortical areas are emerging as potential targets in different diseases [91]. Please rewrite it to be more understandable. What does the author mean by “several other” or “new deep structures”?

Author Response

1) Abstract, raw 39-42 should be rewritten clearly as the aim of the study.

Following the reviewer’s suggestion, we have modified the final part of the abstract to clarify the aim of the manuscript (page 1, line 40-44)

2) Fig 1 what is the source? Ownership of the authors?

Fig 1 is an original figure (ownership of the authors). The figure was drawn by D’Onofrio V. using a vector graphics software (Inkscape) and a graphic tablet (Wacom One).

3) Raw 151-153 – please explain more in detail.

As suggested by the reviewer, the results of the study by Sailer et al [42] were reported more in detail (page 4, line 146-150).

4) Raw 166 – was motor threshold related to resting or active? Kuhn et al. should be written as “Kuhn et al. [Reference number] reported…”

We have now specified that the measure evaluated in the work by Kühn et al. was resting motor threshold and changed the referencing of the study as indicated by the reviewer (page 4, line 171).

5) Raw 171-172, related to studies referenced as 57 and 63, please explain more clearly, whether the stimulation was on on or off?

According to the reviewer’s request, in the revised version of the manuscript we have clarified that in the studies [57] (now reference [58]) and [63] (now reference [56]) SICI was tested with the DBS switched both off and on (page 4, lines 179-180).

6)The acronym usage should be properly checked in the whole manuscript since there are inconsistencies for example: The acronym PAS should be written as the full term in the text (see raw 175, 216). Previously the PAS is introduced as a full term only in the legend of Fig1. The same is for LPF in the text (raw 270), only introduced in Fig 1 as the full term, not in the manuscript text.

Raw 279 is written with the full term “electroencephalographic” and in raw 249, the acronym EEG was introduced.  

We carefully checked for inconsistencies in the manuscript and changed them according to the reviewer’s indications (see comments 11 and 15).

7) Please reduce the number of acronyms in the manuscript.

According to the reviewer suggestion we reduced the number of acronyms in the text. To this aim, the acronyms “IPG” (page 8, lines 357, 362 and 364), “CTC” (page 5, lines 206 and 208) and “MEG” (page 14, line 452) were changed in the corresponding full terms.

8) Raw 180-181, please write in more detail what was done in the Udupa study, and what is the difference between the submitted manuscript and the Udupa?

We have now mentioned that “The major findings of studies applying TMS and DBS alone, as well as the combined application in patients with dystonia, have been discussed in detail in a recent review by Udupa et al. [28]”. (page 5, lines 191-193). Our review is not focused on dystonia by describes more widely the possible applications of TMS and DBS in various neurological conditions, including Parkinson’s disease, essential tremor and epilepsy.

9) Raw 191- “in another study” please rewrite to be more understandable..

In order to make the sentence clearer, “in another study” was changed to “in a previous study by the same group” (page, line 203).

10) Table 1 should be referenced properly in the text, not only in the 3.1.1 section. If  Table 1“covers” other diseases and conditions, then Table 1 referencing should be placed accordingly. It is believed the authors made a mistake with referencing 3.1.1., it should be 2.1.1., and also other paragraphs related to other conditions (dystonia, essential tremor..)?

As suggested by the reviewer, references to Table 1 were added in the text (lines 194, 205, 213, 228 and 273). We also added the legend “Studies using TMS to assess DBS effects on neurophysiological measures” below Table 1. The numbers of subheadings were changed as follows: “2.1.1 Parkinson’s Disease” (page 2, line 100), “2.1.2 Dystonia” (page 4, line 168), “2.1.3 Essential tremor” (page 5, line 195), and “2.1.4 Epilepsy” (page 5, line 209).  

11) Thalamic nuclei should be written with full term or with introducing acronyms, there is nonconsistency in the manuscript (for example, raw 184 versus 197, etc.). It is suggested to use fewer acronyms since it will be too many for the readers to follow.

In the revised version of the manuscript, thalamic nuclei were indicated only with acronyms, and specifically the term “VIM” was used to refer to the ventral intermediate nucleus and “ANT” was introduced to refer to the anterior nucleus of the thalamus (page 5, line 202). We decided to use acronyms instead of full terms because the term “VIM” is used several times in the text (5) and in Table 1 (4).

12) Raw 206, please rewrite to be more understandable “” these studies shed some light..:”

According to the reviewer’s indication, we changed the sentence from “Overall, these studies shed some light on how cortical excitability, connectivity and plasticity are altered in some neurological conditions […]” to “Overall, these studies provided relevant information on abnormalities in cortical excitability, connectivity, and plasticity in several neurological conditions […]” (page 5, line 221).

13) Section 2.2. should be 3.1 and, accordingly, the subheadings? Also, the title should be slightly modified, what does it mean by “study and modulation?

We thank the reviewer for these suggestions. The numbers of all the sections of the manuscript were carefully checked and inconsistences were corrected. Moreover, the title of section 2.2 was changed from “study and modulation of cortical synaptic plasticity” to “modulation of cortical synaptic plasticity” (page 5, line 227).

14) Raw 225 -“has never been” please rewrite to be more acceptable

The sentence has been modified as follows: “To the best of our knowledge, to date, no study applied PAS protocol in PD patients treated with GPi-DBS” (pag6, line 271). 

15) Raw 240- r-TMS, the term is not previously introduced as full term, please reduce the acronym used. The r-TMS is usually used as rTMS.  “has never been” please rewrite to be more acceptable. Raw 242 misses the reference/references? In raw 298, the authors use the term rTMS.

We are sorry for this typing error. In the revised version of the manuscript, we have always used the acronym “rTMS” (which was introduced at line 58) to refer to repetitive transcranial magnetic stimulation. Accordingly, the term “r-TMS” was changed to “rTMS” (page 6, line 288). The missing reference [86] was also added (line 290).

16) Separate paragraph devoted to technical solutions and limitations for the safety application of TMS and DBS should be given, not placing these facts in 2.3. paragraph. No discussion can be found on the difference between the TMS devices (standard TMS with stimulator and EMG device, line navigated TMS and e-field navigated TMS) in usage with DBS in respect of the safety issues or technical compatibilities.

We thank the reviewer for this suggestion. In the revised manuscript, we added a new paragraph (“Safety issues”, page 7 line 336) to better discuss this relevant topic. Regarding the differences between different TMS experimental approaches, and specifically with respect to MRI-guided and E-Field-guided neuronavigation, there is currently no evidence suggesting that they can provide a better stimulation in terms of safety in patients with DBS. We have mentioned this point in the text (page 8, line 364-366).

17) Raw 292 - Kumar et al. should be properly written as Kumar et al. [reference number] according to journal guidelines for text referencing.

We changed the referencing of the study of Kumar et al. according to the journal guidelines (page 8, line 360). We also checked and corrected similar referencing errors throughout the text.

18) Raw 299, “Vagal nerve Stimulator”, why S is the capital, and why V is the capital letter?

The term “vagal nerve stimulation” has been now written in lower case letters (page 13, line 398).

19) Cunic [40] The authors studied the effects of subthalamic nucleus (STN) deep brain stimulation (DBS) on these circuits in 12 patients with PD treated with STN DBS. Please check in Table 1  the number of patients PD (9) and HS 8? The data from HS are served as references/norms for TMS measures right?

In the study by Cunic et al. [40], although twelve patients with Parkinson’s disease were recruited, three of them did not complete the study. Therefore, the results of this work refer to nine parkinsonian patients and eight healthy controls, as reported in Table 1. 

20) It would be suggested to order the studies in Table 1 according to years from more recent to older or from older to more recent studies, and also change the reference numbers accordingly in the text. Or the studies in Table should be ordered as they appeared in the manuscript text (how come study 40 is before 39 in Table 1?).

Following the reviewer’s suggestion, we changed the order of studies reported in Table 1 according to their appearance in the text.

21) 4.1. the margin lining should be placed accordingly to journal guidelines and accordingly to the previous text, here, the paragraphs start as the margins of Table 1.

The margin lining of the text below Table 1 was changed according to the journal guidelines. 

 22) Raw 318 – “..currently under investigation as a treatment for several other neurological disorders, and new deep brain structures as well as cortical areas are emerging as potential targets in different diseases [91]. Please rewrite it to be more understandable. What does the author mean by “several other” or “new deep structures”?

In the revised version of the manuscript, the sentence was changed as follows: “Furthermore, DBS is currently under investigation as a treatment for other neuropsychiatric disorders (e.g., Tourette syndrome, depression and Alzheimer’s disease), and several deep brain structures as well as cortical areas are emerging as new potential targets” (page 13, line 416-419). We hope that the sentence is now more clear to the readership.

Round 2

Reviewer 2 Report

-bolding of Figure and Table text in the manuscript is believed to be redundant, please check the journal guidelines

-raw 98-103 , it would be suggested to slightly rewrite all the future forms such as "we "will first outline" with "we first outline".. "examined", "discussed", without using "we will" 

-Figure 1 as asked by the authors previously, the source needs to be inserted which software was used to draw this picture, ownership

- raw 186-187, "SICI was tested with the DBS switched both off and on, but the researchers found no  change between the two experimental conditions. This sentence needs to be rewritten , instead of "researchers found", it can be written it was found, or study found...

-Raw 270 , it is believed that this sentence misses the reference or more clearer explanation : "To date, only Paired Associative Stimulation (PAS) was i applied in DBS patients to assess changes in cortical synaptic plasticity"

-raw 349 "andmore" this should be divided, and the whole manuscript should be checked by a native speaker for the grammatical and writing style. Raw 352[90]recently...; [106]to date ...etc.

-  "100% of mean stimulator output". What does it mean "mean", do you mean maximal stimulator output? It is written on several spots

-Raw 372, please be clearer when explaining "standard TMS" or E-field TMS" This is not yet clear to the readers.

-Raw 376-378, the Cunic and Chen should also have reference numbers d when being also cited under the table

-What the empty dot means above "not clearly reported"? it is not explained in the table legend

-Raw 452-453 , "to date no study has been performed in these patients" Please state more clearly which studies have not been done.

Author Response

Comments to the authors

1) bolding of Figure and Table text in the manuscript is believed to be redundant, please check the journal guidelines

According to the journal guidelines, in the revised version of the manuscript we used light typing when referencing “Figure 1” and “Table 1” in the text (lines 83, 123, 178, 193, 201, 216 and 290).

2) raw 98-103 , it would be suggested to slightly rewrite all the future forms such as "we "will first outline" with "we first outline".. "examined", "discussed", without using "we will" 

Lines 95-100 were changed according to the reviewer’s suggestion. 

3) Figure 1 as asked by the authors previously, the source needs to be inserted which software was used to draw this picture, ownership

Following the reviewer’s indication, in the legend of Figure 1, we added the graphic software used to create the figure (Inkscape version 1.2.2), and we also specified that the figure was drawn by the V.D.

4) raw 186-187, "SICI was tested with the DBS switched both off and on, but the researchers found no  change between the two experimental conditions. This sentence needs to be rewritten , instead of "researchers found", it can be written it was found, or study found...

In the revised version of the manuscript, the sentence was changed as follows: “SICI was tested with the DBS switched both off and on, but no change was found between the two experimental conditions” (lines 183-184).

3) Raw 270 , it is believed that this sentence misses the reference or more clearer explanation : "To date, only Paired Associative Stimulation (PAS) was i applied in DBS patients to assess changes in cortical synaptic plasticity"

According to the reviewer’s indication, we have now specified that “PAS is the only TMS protocol applied in DBS patients to assess changes in cortical synaptic plasticity”, and we added references [26] and [28] corresponding to previous works supporting the statement (lines 267-268).

4) raw 349 "andmore" this should be divided, and the whole manuscript should be checked by a native speaker for the grammatical and writing style. Raw 352[90]recently...; [106]to date ...etc.

Following the reviewer’s indication, the manuscript was revised again for the English language by a native English speaker.

5) "100% of mean stimulator output". What does it mean "mean", do you mean maximal stimulator output? It is written on several spots

As suggested by the reviewer, we replaced the incorrect term “mean stimulator output” with “maximum stimulator output” (lines 354, 377 and 386).

6) Raw 372, please be clearer when explaining "standard TMS" or E-field TMS" This is not yet clear to the readers.

As pointed out by the reviewer, we changed the term “standard” with “non-navigated” when referring to TMS approaches performed without neuronavigation techniques (line 370).

7) Raw 376-378, the Cunic and Chen should also have reference numbers d when being also cited under the table

According to the reviewer’s suggestion, we added reference numbers to the studies cited in the legend of Table 1 (lines 417-418).

8) What the empty dot means above "not clearly reported"? it is not explained in the table legend

As pointed out by the reviewer, we added the missing empty dot (°) in the legend of Table 1 before the sentence “The authors do not specify patients’ medication, but report that medication was not changed during the study” (line 419).

9) Raw 452-453 , "to date no study has been performed in these patients" Please state more clearly which studies have not been done.

In the revised version of the manuscript, we have now clarified that “studies combining TMS with DBS have never been performed in these patients” (lines 452-453).